# The Contribution of Autophagy and LncRNAs to MYC-Driven Gene Regulatory Networks in Cancers

**DOI:** 10.3390/ijms22168527

**Published:** 2021-08-08

**Authors:** Leila Jahangiri, Perla Pucci, Tala Ishola, Ricky M. Trigg, John A. Williams, Joao Pereira, Megan L. Cavanagh, Suzanne D. Turner, Georgios V. Gkoutos, Loukia Tsaprouni

**Affiliations:** 1Department of Life Sciences, Birmingham City University, Birmingham B15 3TN, UK; leila.jahangiri@bcu.ac.uk (L.J.); tala.ishola@bcu.ac.uk (T.I.); megan.cavanagh@mail.bcu.ac.uk (M.L.C.); 2Division of Cellular and Molecular Pathology, Department of Pathology, University of Cambridge, Cambridge CB2 0QQ, UK; pp504@cam.ac.uk (P.P.); sdt36@cam.ac.uk (S.D.T.); 3Department of Functional Genomics, GlaxoSmithKline, Stevenage SG1 2NY, UK; ricky.m.trigg@gsk.com; 4Institute of Translational Medicine, University Hospitals Birmingham NHS Foundation Trust, Birmingham B15 2TH, UK; j.a.williams@bham.ac.uk (J.A.W.); g.gkoutos@bham.ac.uk (G.V.G.); 5Institute of Cancer and Genomic Sciences, College of Medical and Dental Sciences, University of Birmingham, Birmingham B15 2SY, UK; 6Department of Neurology, Massachusetts General Hospital, Harvard Medical School, Boston, MA 02114, USA; jdpereira@cantab.net; 7CEITEC, Masaryk University, 625 00 Brno, Czech Republic; 8Mammalian Genetics Unit, Medical Research Council Harwell Institute, Oxfordshire OX11 0RD, UK; 9MRC Health Data Research, Birmingham B15 2TT, UK; 10NIHR Experimental Cancer Medicine Centre, Birmingham B15 2TT, UK; 11NIHR Surgical Reconstruction and Microbiology Research Centre, Birmingham B15 2TT, UK; 12NIHR Biomedical Research Centre, Birmingham B15 2TT, UK

**Keywords:** MYC, gene regulatory networks (GRNs), autophagy, lncRNAs

## Abstract

MYC is a target of the Wnt signalling pathway and governs numerous cellular and developmental programmes hijacked in cancers. The amplification of MYC is a frequently occurring genetic alteration in cancer genomes, and this transcription factor is implicated in metabolic reprogramming, cell death, and angiogenesis in cancers. In this review, we analyse MYC gene networks in solid cancers. We investigate the interaction of MYC with long non-coding RNAs (lncRNAs). Furthermore, we investigate the role of MYC regulatory networks in inducing changes to cellular processes, including autophagy and mitophagy. Finally, we review the interaction and mutual regulation between MYC and lncRNAs, and autophagic processes and analyse these networks as unexplored areas of targeting and manipulation for therapeutic gain in MYC-driven malignancies.

## 1. Introduction to MYC Transcription Factors and Their Roles in Cancer

The MYC family of proteins are basic helix-loop-helix leucine zipper (bHLHZip) transcription factors (TFs) under tight transcriptional regulation. The MYC family includes c-MYC (MYC), the first member identified in humans; MYCL; and MYCN [1], encoded by *c-MYC*, *L-MYC*, and *N-MYC *(*MYCN*) genes, respectively [2]. MYC TFs can be transactivated by forming heterodimers with their partner protein, MAX, which then bind to E-box motifs and, in association with the transcription machinery, trigger the regulation of proliferation, metabolism, differentiation, cell cycle, apoptosis, DNA damage, angiogenesis, protein synthesis, and mitochondrial function [3,4,5]. However, the low specificity of MYC binding to E-box motifs can lead to the activation of alternative pathways and changes to metabolic states [3,4,5]. Furthermore, MYC sits at the crossroads of numerous signalling pathways where it can act as an early response mediator, and precise regulation of MYC expression is critical for maintaining the balance between proliferative and differentiated cellular states [6]. MYC TFs are downstream of the Wnt signalling pathway and can be aberrantly expressed due to mutations in β-catenin and APC and modulation by non-coding RNAs [7,8]. MYC expression is also dependent on other signalling pathways, including PI3K/AKT/mTOR, Notch, TGF-β/SMADs, and growth factor-induced pathways [9,10,11]. In Figure 1, we highlighted the roles of Wnt, EGF, and TGF-β signalling pathways in MYC regulation.

MYC TFs are expressed in normal embryogenesis and play integral roles in influencing normal developmental processes, such as brain development, including the cerebellum [12]. Furthermore, MYC inactivation may be a prerequisite for terminal differentiation, lineage commitment, and cellular quiescence [13,14]. For instance, in haematopoietic progenitor cells and skin stem cells, MYC can induce terminal differentiation and lineage commitment [15,16].

In cancers, however, MYC has also been shown to interact with other oncogenes, such as RAS, in numerous cancers [17]. MYC activation can, when in association with other mutagenic events, drive tumourigenesis. For instance, *MYC* expression driven by different enhancers, in association with other mutations, led to leukaemia formation in various B-cell developmental stages [8]. Moreover, alterations to MYC expression and function are found in multiple malignancies, including cancers of the central and peripheral nervous system such as glioblastoma multiforme, and neuroblastoma (NB) [18,19]. In many of these cancers, MYC hijacks cellular and molecular programmes through an extensive network of target genes, effectors, regulators, and signalling pathways. MYC has also been implicated in the reprogramming of metabolic states and in promoting angiogenesis, aggressive behaviour, and metastasis in cancers [20,21,22,23]. Table 1 shows examples of genes associated with MYC TFs in a panel of cancers.

In addition to MYC TFs’ roles in cell proliferation, metabolism, and angiogenesis, these TFs may also be involved in lineage plasticity in cancers. Cell plasticity is defined as the ability of a cell to alter its phenotype without genetic mutation in response to environmental factors. It is considered an emerging mechanism of tumour evolution and treatment resistance in cancers [24]. For instance, MYCN-driven, castration-dependent prostate cancer may progress towards a neuroendocrine fate through lineage plasticity and epigenetic reprogramming, reducing therapy efficacy [24].

MYC genes and proteins also interact with long non-coding RNAs (lncRNAs) by regulating various cellular pathways and processes in cancer. Notably, lncRNAs have recently been revealed to target *MYC* genes and proteins by regulating either MYC transcription, protein stabilisation, activity, or other molecules involved in MYC expression and activity [32,33]. As we will discuss later in this review, the interaction between MYC and lncRNAs can be complex and fundamental to cancer progression via the regulation of numerous processes, such as proliferation, survival, migration, and invasion.

MYC TFs have also been found to have a central role in cancer by inducing autophagy. For instance, cells following transformation induced by MYC activated autophagy due to mitochondrial stress, inhibiting tumour growth in vitro and in vivo [34]. This study demonstrated that MYC-mediated transformation sensitised cells to autophagy induction by the linamarase/linamarin/glucose oxidase system (lis/lin/GO), which activated the AMPK pathway and upregulated autophagy genes [34,35,36]. These studies revealed a central role for MYC-mediated cellular functions, signalling, and metabolic pathways.

In this review, we address the role of MYC as a transforming oncogene in multiple cancers and how MYC is influenced by lncRNAs and may be implicated in crucial pro-survival cellular processes such as autophagy. Finally, we discuss both these aspects of MYC pathology to identify novel therapeutic strategies that may be implemented to effectively establish MYC as a valid candidate for onco-therapy.

## 2. Cancers with MYC-Related Functions in Their Pathogenesis

### 2.1. Prostate Cancer

MYCN is not usually expressed in the epithelial lineage that gives rise to prostate cancer. However, its aberrant expression can, in part, drive neuroendocrine prostate features in advanced prostate cancer by epigenetic reprogramming [24]. A study aimed at understanding the MYCN-triggered epigenetic reprogramming of prostate cancer to neuroendocrine prostate fate, the cistrome, and the transcriptome of these tumours were investigated in vitro, in vivo, and in patient-derived organoids. Genetically modified mice, *Pb-Cre+/− Ptenfl/fl LSL-MYCN+/+*, develop prostate tumours in 100% of offspring compared with littermates that lack MYCN expression (*Pb-Cre+/− Ptenfl/fl LSL-MYCN−/−*). In response to castration, *Pb-Cre+/− Ptenfl/fl LSL-MYCN+/+* mice developed invasive tumours with metastatic features including epithelial-to-mesenchymal transition (EMT) (e.g., indicated by vimentin expression) and the expression of neural markers (e.g., NCAM1) [24]. Furthermore, transcriptional profiling of MYCN-expressing 22Rv1 prostate cancer xenograft models in castrated and non-castrated states revealed that castration led to an increase in expression of neural lineage genes (e.g., *SOX11*). This observation was also confirmed in LNCaP-MYCN overexpressing cell lines, in which androgen absence led to the differential expression of over 40% of MYCN target genes (including stem cell and neural lineage differentiation markers). Contrastingly, in the presence of androgen, androgen receptor (AR)-related genes were preferentially expressed [24]. Consistent with the acquisition of an alternative identity, for LNCaP-MYCN expressing cell lines, the MYCN-driven cistrome was revealed to be AR-dependent. In the presence of androgen, MYCN co-occupied genomic loci with targets of AR, including *FOXA1* and *HOXB13*. In the absence of androgen, the MYCN cistrome was diverted towards promoters of neural stem cell (NSC) genes. In conclusion, MYCN-driven prostate tumours displayed greater aggressiveness, with MYCN driving transcriptional programmes in advanced stages of prostate cancer [24,37].

### 2.2. Pancreatic Ductal Adenocarcinoma

In pancreatic ductal adenocarcinoma (PDAC) originating from exocrine acinar cells, acinar to ductal metaplasia and progression to high-grade neoplasia were instigated upon acquiring *Kras* mutations [25]. *Kras* mutant tumours maintained metabolic reprogramming mediated by Myc and increased glycolytic flux and glutamine metabolism to meet the energetic demands of elevated proliferation and redox maintenance, respectively [37]. Yap, a transcriptional regulator, acted as a driver of *Kras*-mediated PDAC initiation [38], while the roles of Myc, Yap, and *Kras* mutations in the maintenance of PDAC were investigated using conditional transgenic mouse models. *Kras^G12D^* and *Yap* were sequentially activated or deleted using Flp-FRT and tamoxifen-inducible Cre-lox systems, respectively (i.e., *FSF-Kras^G12D/+^*, *R26^FSF-CreER/dual^*, *Yap^flox/flox^*, and *pdx1-Flp*) [25]. From a metabolic standpoint, the co-expression of Yap and Myc promoted metabolic gene expression, leading to tumour growth, maintenance, and survival in PDAC [25]. Furthermore, in partnership with TEA domain (TEAD) TFs, Yap promoted *Myc* transcription and cooperated with Myc in regulating metabolic gene expression. The ablation of Yap led to a metabolic crisis, regression of tumours at early stages, and cell death. Interestingly, in the absence of Yap, Myc expression levels were upregulated by Sox2, resulting in a reversal of the inhibitory effect of Yap deletion on cell proliferation and metabolic gene regulation. This compensatory effect ultimately led to the re-differentiation of neoplastic ductal cells to acinar cells with associated pancreatic enzyme production, highlighting the complexity of MYC-driven metabolic networks [25].

### 2.3. Medulloblastoma

Medulloblastoma is an aggressive paediatric brain tumour classified into the WNT, SHH, group 3, and 4 subtypes. In subgroup 3 of this cancer, somatic *MYC* alterations and amplifications have been reported, although it remains unclear whether these alterations alone drive tumourigenesis [26]. In a study aimed at understanding the contribution of MYC to malignant transformation, lentiviral *MYC* was transduced into unsorted mouse cerebellum cells, which were subsequently orthotopically transplanted into NOD SCID gamma (NSG) mice, leading to tumour formation. This study also revealed that Sox2 was required for tumour initiation since the co-expression of endogenous *Sox2* with transduced *MYC* was sufficient to lead to tumourigenesis in vivo [26]. This study also demonstrated that the *Sox2*^+^ cell population expressing Aldh1a1, a protein associated with a cancer stem cell phenotype, generated tumours in vivo, while this capacity decreased with lower or absent expression of Aldh1a1 [26]. Additionally, the expression of lactate dehydrogenase A (*LDHA*) positively correlated with MYC expression levels indicative of an unfavourable prognosis in this medulloblastoma subtype [26]. The inhibition of LDHA in vivo led to reduced growth of MYC-driven tumours but not MYC-independent tumours [26]. In conclusion, this study revealed that MYC-induced transformation of Sox2^+^/Aldh1a1^high^ cells led to tumour growth in the postnatal cerebellum in vivo, with characteristics of type 3 medulloblastoma [26].

### 2.4. Neuroblastoma

Risk stratification for NBs is dependent on multiple variables, including MYCN amplification status and age [39]. Of these, MYCN amplification, seen in more than 50% of high-risk NBs, is the strongest indicator of poor clinical prognosis. The mechanism by which MYCN influences NB development was investigated in a study that analysed the transcriptomic profile, specifically miRNA and mRNA interactions implicated in tumourigenesis, of premalignant sympathetic ganglia and tumours from tyrosine hydroxylase promoter-driven MYCN (*TH-MYCN^+/+^*) transgenic mice [28]. For instance, *mir-204* was shown to suppress a network of oncogenes associated with MYCN, while *mir-204* and MYCN mutually repressed each other’s expression by binding to the corresponding promoter, in association with relevant transcriptional machinery and repressors [28]. Through this negative repression mechanism, the insertion of *miR-204* mimics into MYCN amplified NB cell lines led to a reduced colony-forming capacity. Accordingly, *miR-204* was identified as a tumour suppressor and a putative therapeutic agent, since in mouse xenograft experiments, MYCN-amplified cells transduced with an inducible *miR-204* construct showed delayed tumourigenesis [28].

### 2.5. Colorectal Cancer

Deregulation of the Wnt-APC pathway can lead to increased levels of MYC in colorectal cancer (CRC) [7]. The role of MYC in metabolic reprogramming of CRC was investigated in a study by Satoh and colleagues in which they compared multi-omics profiles across 275 matched tumour and normal patient tissues. The upregulation of MYC was detected at all stages of cancer progression and resulted in the aberrant regulation of 231 metabolic genes involved in pathways including MAPK signalling, pentose phosphate, and fatty acid metabolism [7]. Interestingly, the most remarkable changes to these metabolic genes, or a metabolic shift, were detected at the adenoma stage and maintained in later stages of the tumourigenic process. Interestingly, the knockdown of *MYC* in vitro led to a reversal of the altered metabolism and reduced cell growth. This study provided evidence that MYC, as a regulator of metabolic processes, governed the metabolic reprogramming of CRC by regulating metabolic reactions and a multitude of genes involved in metabolism, such as *CAD*, *UMPS*, and *CTPS* [7]. This finding may impact diagnostic studies.

### 2.6. Rhabdomyosarcoma

Over 40% of alveolar Rhabdomyosarcoma (RMS) show MYCN amplification associated with improved overall survival. MYCN expression is also characteristic of embryonal RMS, although gene amplification is not a feature in this subtype [40,41,42]. In a study focused on the mechanisms behind RMS tumourigenesis, immortalised human myoblasts expressing constitutive MYCN and doxycycline-inducible PAX3-FOXO1 (a fusion protein) were grafted orthotopically to immunodeficient mice. Food supplementation with doxycycline led to the induction of PAX3-FOXO1 and the formation of RMS tumours. Cells transformed exclusively with PAX3-FOXO1 generated tumours after a longer period of time than those also expressing MYCN. These results suggest cooperation between these two TFs in the early stages of tumourigenesis, leading to an increase in proliferation and the inhibition of the myogenic differentiation programme [29]. Interestingly, the withdrawal of doxycycline from animal diets after tumour formation led to tumour regression followed by tumour recurrence, suggesting that, while RMS tumours are dependent on PAX3-FOXO1, these may relapse through PAX3-FOXO1-independent pathways [29].

### 2.7. Non-Small Cell Lung Cancer

MYC and KRAS act as downstream conduits for other oncogenic drivers, enhancing the process of tumourigenesis in lung adenomas and their transformation to invasive and proliferative adenocarcinoma of the lung [30,31]. A study by Soucek and colleagues, focusing on an endogenous Myc-switchable *Kras^G12D^* (*LSL-Kras*, *TRE-Omomyc*, and *CMVrtTA*) mouse model of lung adenocarcinoma showed that *Kras^G12D^* expressing mice developed bronchiolar hyperplasia [31]. The Kras-dependent bronchiolar hyperplasia regressed with *Omomyc* expression, suggestive of a role for Myc in maintaining *Kras^G12D^*-driven lung tumours. Mechanistically, *Omomyc* expression inhibited cellular proliferation by downregulating Myc expression. Interestingly, Myc inhibition in normal, healthy tissues with a rapid turnover, such as the epidermis, testis, and intestinal tract displayed reduced proliferation. Despite these effects, the systematic suppression of Myc was tolerated since the reversal of Myc inhibition led to rapid recovery of the affected tissues [31].

In a follow-up study using *LSL-Kras^G12D^Rosa26-LSL-MycERT2* transgenic mice, with constitutive expression of *Kras^G12D^* and inducible expression of *MycER^T2^*, it was proposed that, while the *Kras^G12D^* mutation alone was sufficient to induce preneoplastic lesions, the additional activation of Myc led to rapid acceleration of tumourigenesis, hence transforming indolent tumours to adenocarcinomas [30]. This group also investigated the effect of *MycER^T2^* activation on angiogenesis and the tumour microenvironment, revealing that Myc governed processes such as inflammation and angiogenesis through its non-cell-autonomous effectors, IL-23 and CCL9, which induce microenvironment alterations. Consistently, the blockage of CCL9 alone led to the extensive inhibition of macrophage influx mediated by Myc, loss of T cells, and the prevention of new blood vessel formation. In contrast, the inhibition of IL-23 led to tumour cell death by apoptosis; a reduction in proliferation; and the recruitment of B, T, and NK cells [30]. Ultimately, this study pointed towards the marked capacity of Myc to suppress innate and adaptive immunity in tissues and to induce adenocarcinomas from indolent tumours.

In summary, we reviewed MYC-driven regulatory networks in multiple cancers and highlighted potential novel therapeutic axes involved in each specific cancer type. Due to the difficulty in directly targeting MYC for therapy, these studies were invaluable and could lay the groundwork for successful MYC-focused treatment strategies.

## 3. MYC and Long Non-Coding RNAs (LncRNAs) in Cancer

### 3.1. LncRNAs in Cancer as Novel Biomarkers and Therapeutic Targets

LncRNAs have been shown to interact with TFs, including MYC, often via direct regulation of their expression, protein stabilisation, and activity or via indirect regulation of other molecules involved in their expression and activity [32,33,43]. LncRNAs are non-coding RNA (ncRNA) molecules longer than 200 nucleotides and are the most abundant class of non-coding transcripts. LncRNAs are aberrantly expressed in several malignancies, and both the lncRNAs and the mechanisms associated with the complex structures they can form, allowing for their interaction with a large number and variety of molecules, have been extensively studied in the context of cancer biology and treatment [33]. Notably, lncRNAs are often expressed in a tissue- and disease-specific manner with critical functions in several cancers and have great potential as novel biomarkers and therapeutic targets in cancer [44,45,46]. Consistently, lncRNAs have been isolated from patient’s biological fluids, either free or packed into exosomes [47,48]. Clinical trials are currently recruiting patients for lncRNA-based cancer diagnostics (URL: https://clinicaltrials.gov/ct2/show/NCT03830619) (accessed 15 February 2019); a trial focused on using serum exosomal lncRNAs as putative biomarkers in lung cancer diagnostics. Other lncRNAs have already been approved for clinical use; for example, PCA3 is currently used in the diagnosis of prostate cancer [49]. Additionally, most lncRNAs are upregulated in specific oncogenic pathways, and their inhibition could act alone or in combination with other treatments to improve patient response. Several methods of lncRNA inhibition include antisense oligonucleotides (ASOs), single-stranded DNA polymers, locked nucleic acids (LNA), and morpholinos (Table 2) [50,51,52,53,54,55,56].

### 3.2. Oncogenic IncRNAs Promote MYC Expression and Activity in Cancer

Several studies have focused on the roles of lncRNAs in MYC regulation in cancer; in recent years, novel mechanisms and interactions that are of clinical relevance have been characterised. One notable mechanism is the mutual regulation of lncRNAs and MYC via intricate feedback loop mechanisms. LncRNAs are often upregulated in cancer and can promote oncogenic pathways and phenotypes. For example, a regulatory network modulated by *MALAT1* in thyroid cancer has shown that this lncRNA acts as a competing endogenous RNA (ceRNA) by sequestering *miR-204* (Figure 2A) [32]. In Figure 2A, *MALAT1* and *miR-204* have been referred to as lncRNA and miRNA, respectively. Specifically, this study revealed that, in thyroid cancer, *MALAT-1* and insulin-like growth factor 2 mRNA binding protein 2 (IGF2BP2), a member of RNA binding proteins, were expressed while *mir-204* was expressed poorly. IGF2BP2 was identified as a target of *mir-204* and the regulatory loop comprising these players was identified as follows: *MALAT1* selectively bound to *mir-204*, preventing *mir-204* from binding IGF2BP2 and leading to IGF2BP2 upregulation. As a result, IGF2BP2 could recognise the N6-methyl-adenosine (m6A) modification of *MYC* RNA and could consequently increase its expression. This regulatory network resulted in increased proliferation, migration, and invasion of thyroid cancer cells [32]. Hence, targeting lncRNAs may simultaneously affect numerous downstream oncogenic pathways, including MYC, representing a valid therapeutic approach.

*Linc00485* is another example of an oncogenic lncRNA for which expression is upregulated in human lung cancer and is associated with metastasis and relapse [57]. Notably, *linc00485* was associated with the tumour-node-metastasis (TNM) stage in this cancer, and *linc00485* overexpression increased the proliferation, migration, and invasiveness of lung cancer cells by directly sequestering *miR-298*, which targeted MYC. The significance of the *linc00485*/*miR-298*/MYC axis in vivo, demonstrated using xenotransplantation of lung cancer cell lines including A549, revealed that *linc00485* silencing using short hairpin lincRNA resulted in reduced cancer cell proliferation. Collectively, this study showed a crucial regulatory link involving MYC, miRNAs, and lncRNAs while offering a promising therapeutic strategy (Table 2) [57].

### 3.3. LncRNAs Repress MYC under Certain Conditions

LncRNAs are not exclusively oncogenes and can also act as tumour suppressors in specific circumstances. An example is *linc00261*, which acts as a tumour suppressor in pancreatic cancer via transcriptional regulation of *MYC* expression. *Linc00261* directly binds to the bromodomain of the transcriptional co-factor p300/CBP (a member of the transcriptional co-activator family), preventing its recruitment to the promoter region of *MYC*, thereby epigenetically repressing *MYC* expression (Figure 2B, *linc0026* is lincRNA) [58]. Finally, *linc0026* overexpression inhibited pancreatic cancer cell proliferation, migration, and metastasis in vitro (Table 2) [58].

An alternative mechanism of *linc00261* activity in pancreatic cancer is via targeting of *miR-222-3*, leading to the modulation of MYC expression both via miRNA regulation and sequestration of insulin-like growth factor 2 mRNA binding protein 1 (IGF2BP1). Mechanistically, *linc00261* was downregulated by methylation of the CpG islands (regions of DNA with greater CG content) associated with its promoter, and the enhancer of Zeste 2 Polycomb repressive complex 2 subunit (EZH2) mediated the methylation of lysine 27 of histone 3 (H3K27) (also known as permissive histone marks) deposited to its promoter region. This suggests a complex interaction between lncRNAs and epigenetics in the regulation of oncogenes such as MYC [58,64].

Another example, *PVT1* lncRNA, was a regulator of the transcription of key oncogenic pathways, such as TGFβ/SMAD and Wnt/β-catenin, acting as an enhancer for MYC [65]. Specifically, the *PVT1* locus impacted the expression of essential genes in both pathways while the aberrant methylation of *PVT1* lncRNA also led to alterations in *MYC* expression. Therefore, it is not surprising that the *PVT1* lncRNA expression was associated with a poor prognosis for CRC patients [65].

On the contrary, in lung adenocarcinoma, the *PVT1b* isoform is expressed downstream of MYC and is induced by DNA damage and oncogenic signals, thus acting as a tumour suppressor [66]. Once transcribed, *PVT1b* accumulated near the MYC transcription start site, contributing to the suppression of MYC expression in cis, a mechanism triggered by P53 [66]. It follows that *PVT1b* inhibition resulted in an escalation of MYC expression and activity, leading to an increase in cell proliferation; indeed, loss of *PVT1b* in vivo promoted tumour growth [66]. In conclusion, the P53-induced lncRNA-mediated changes in transcription may allow for rapid adaptation and response to cellular stress.

### 3.4. Mutual Regulation of IncRNAs and MYC via Feedback Loops

LncRNA-MYC feedback loop mechanisms have been studied in different malignancies with biological and clinical relevance to cancer phenotypes. An interesting example may be metastasis suppressor 1 (*MTSS1*) and its lncRNA. In pancreatic cancer, the lncRNA *MTSS1-AS* upregulated its sense gene, metastasis suppressor 1 (*MTSS1*), by acting as a scaffold/decoy between E3 ubiquitin-protein ligase STIP1 homology and U-box containing protein 1 (STUB1), which earmarks proteins for proteasomal degradation and the transcription regulator myeloid zinc finger 1 (MZF1). MZF1 inhibited *MTSS1* expression by binding its promoter (Figure 3A, *MTSS1-AS*, *MTSS1*, and MZF1 are lncRNAs encoded by gene *A-AS*, gene *A*, and regulatory protein, respectively), and *MTSS1-AS*-induced ubiquitination-mediated degradation of MZF1 leads to the upregulation of *MTSS1* [67]. Furthermore, extracellular acidity reduced *MTSS1-AS* levels, thereby stabilising MZF1 and promoting metastasis [67].

MYC regulation was upstream of this pathway, and MYC binds the initiator elements of the *MTSS1-AS* promoter, thereby inhibiting lncRNA transcription in association with relevant transcriptional machinery [67]. In turn, *MTSS1-AS* repressed the MZF1-mediated transcription of *MYC*, thereby forming a negative feedback loop between *MTSS1-AS* and MYC in acidic pancreatic cancer cells, with overexpression or knockdown of *MTSS1-AS* leading to the inhibition or promotion of *MYC* expression, respectively (Figure 3B, *MTSS1-AS*, *MTSS1*, and MZF1 are lncRNAs encoded by gene *A-AS*, gene *A*, and regulatory protein, respectively).

Finally, lncRNA-MYC feedback loops included LPP antisense RNA-2 (*LPP-AS2*), a lncRNA that was found to be upregulated in glioma. Specifically, *LPP-AS2* sequestered *mir-7-5p*, thereby increasing the expression of EGFR and the downstream PI3K/AKT/MYC axis, promoting the transcription of MYC, via this positive feedback loop. In turn, MYC bound directly to the promoter of *LPP-AS2* [59]. Depletion of the *LPP-AS2/miR-7-5p*/EGFR/MYC axis reduced glioma cell proliferation and invasion, and triggered apoptosis. In vivo *LPP-AS2* knockdown inhibited tumour growth, whereas *LPP-AS2* overexpression had the opposite effect [59]. The oncogenic activity of *LPP-AS2* in this positive feedback loop may be a target for therapeutic interventions (Table 2) [59].

To summarise, we reviewed the regulatory complexity surrounding the circuits comprising of MYC and various lncRNAs, miRNA, and signalling pathways. LncRNAs can act as oncogenes by increasing MYC expression and activity or by inhibiting MYC expression. This regulation of MYC function is dependent on multiple positive and negative feedback loops. The enhanced understanding of these regulatory networks could pave the way to the effective targeting of MYC-driven regulatory networks and could introduce lncRNAs as novel biomarkers in cancers and putative therapeutic targets.

## 4. MYC and Autophagy

### 4.1. MYC, Autophagy, and Cellular and Molecular Processes in Cancers

Macro-autophagy, hereafter referred to as autophagy, is a major degradation pathway for many organelles, toxic aggregates, and long-lived proteins. In this process, fractions of the cytoplasm are engulfed in a double-membrane structure, also known as an autophagosome. This structure is then fused with lysosomes, allowing for the degradation of its content. *MYC*, along with other genes such as *mTOR, Beclin-1, p53, PTEN, p62, MIF, HMGB1, RAC3, SRC3, NF-2, MEG3, LAPTM4B*, and *BRAF* play essential roles in autophagy-induced tumourigenesis and drug resistance, a new focus of drug development efforts [68]. Toh and colleagues showed that MYC was involved in autophagosome formation in the early stages of autophagy rather than in its degradation and that MYC-driven modulation of autophagy occurs via the JNK1-Bcl2 pathway and ROS (Figure 4). The siRNA-mediated knockdown of *MYC* inhibited autophagy in human embryonic kidney (HEK293) cells, resulting in an accumulation of the autophagy substrate p62 (SQSTM1/sequestome1, an autophagy receptor, and signalling adapter). Hence, MYC inhibition led to defective autophagosome formation and reduced the clearance of essential autophagy substrates [69]. This study pointed towards the additional benefit of MYC modulation on autophagic processes, especially in autophagy-induced therapy resistance.

Moreover, MYC-induced alterations in autophagy can impact tumourigenesis, whereby the dysfunction of MYC-mediated autophagy contributed to non-small cell lung cancer (NSCLC) development [60]. In NSCLC, inhibiting MYC/*miR-150* expression greatly reduced cell growth in vitro and in vivo (Table 2) [60]. In *miR-150* overexpressing cell lines, a dysfunction in autophagic flux was observed, as shown by higher numbers of autophagosomes and lower numbers of autolysosomes. *miR-150* inhibited the fusion of autophagosomes and lysosomes by directly repressing *EPG5*, a gene essential for autophagosomal maturation, consequently promoting NSCLC development. Since autophagy deficiency caused the accumulation of damaged mitochondria, it was not surprising that ROS levels were increased in *miR-150* overexpressing A549 and H1299 NSCLC cells [60].

In parallel with the aforementioned reports, numerous studies have alluded to the association of multiple factors with the autophagic processes that may affect MYC protein stabilisation. Noteworthily, MYC protein stability is defined by the interaction between two phosphorylation sites, serine 62 and threonine 58 (by ERK and glycogen synthase kinase-3β, GSK3β, respectively). Serine 62 phosphorylation leads to MYC stabilisation but promotes the phosphorylation of threonine 58, destabilising the protein [70]. Here, we focus on two examples.

In osteosarcoma, polo-like kinase 1 (PLK1, a serine/threonine protein kinase) and *MYC* promote cell proliferation through the autophagic pathway [71]. PLK1, a protein kinase, is a major driver of proliferation and growth that correlated with poor prognosis when expressed in this cancer. The knockdown of *MYC* led to decreased LC3-II/LC3-I (standard markers of autophagosomes) and autophagy-related protein 7 (Atg7) as well as defects in autolysosomal degradation. Similarly, the knockdown of *PLK-1* resulted in significantly decreased autophagy markers, such as LC3-II/LC3-I and Atg5, in parallel with SQSTM1 accumulation and defects in the autolysosomal pathways. Moreover, PLK-1 contributed to *MYC* protein stabilisation while PLK1 inhibition led to a significant loss of MYC abundance. Consequently, this caused a marked delay in xenograft tumour growth in mice treated with the PLK-1 inhibitor BI2536, with a lower mean tumour volume compared with the control group [71].

Another protein with a role in autophagy initiation is AMBRA1, a scaffold protein that is a downstream target of mTOR, a major regulator of autophagy. This protein was found to play a tumour suppressor role by promoting MYC dephosphorylation and degradation in lung and breast cancer cell lines. Disruption of the *Ambra1 locus* induced MYC hyperphosphorylation, leading to tumour hyperproliferation and tumourigenesis. Moreover, when mTOR was inhibited, AMBRA1 enhanced the interaction between MYC and its phosphatase PP2A, consequently reducing cell proliferation (Table 2) [61]. From these studies, it is possible to put forth PLK1 and AMBRA1 as valuable candidates for therapies aimed at targeting MYC protein stability.

In addition to the examples above, MYC may also be associated with the pathway connecting the endoplasmic reticulum (ER) stress and autophagy. Hart and colleagues showed that ER stress-mediated autophagy stimulated *MYC*-dependent transformation and tumour growth. Specifically, they showed that MYC and MYCN activated the PERK/eIF2α/ATF4 arm of the unfolded protein response (UPR) (one of the three arms of UPR) in the P493-6 human lymphoblastoid cell line and mouse embryonic fibroblast (MEF) cells, resulting in increased cell survival via the induction of autophagy (Table 2) [62]. The inhibition of PERK led to reduced MYC-induced autophagy and a significant decrease in tumourigenesis, while the inhibition of autophagy increased the level of apoptosis in an MYC-dependent manner. Blocking ER stress was sufficient to restore normal levels of protein synthesis and consequently reduced autophagy. This link between UPR, ER stress, and autophagy may provide an attractive therapeutic target [62].

### 4.2. Chaperone-Mediated Autophagy (CMA) and Mitophagy Link with MYC

MYC has also been linked with another form of autophagy, chaperone-mediated autophagy (CMA), a selective form of protein lysosomal degradation (Table 2). In most cancers, CMA is upregulated and is essential for tumour growth [72]. Kon and colleagues revealed a novel tumour-suppressor role for CMA in MEF cells, where CMA inhibited the oncogenic activity of MYC by promoting its proteasomal degradation. Mechanistically, CMA triggered MYC destabilisation by regulating CIP2A degradation. CIP2A inhibited a phosphatase that dephosphorylated MYC at Ser62, preventing the ubiquitination and subsequent degradation of MYC (Table 2). Naturally, blocking CMA led to reduced degradation of CIP2A and increased stability of MYC, acting a as potential preventative strategy in cancer initiation (Figure 5, CIP2A is referred to as a regulatory protein) [63].

Consistently, other studies have shown a link between MYC and mitophagy, a process of selective degradation of mitochondria by autophagy following stress or damage [73]. The mild and sustained hydrogen peroxide treatment induced Parkin-mediated mitophagy and reduced the accumulation of GSK3β in the nucleus, thus decreasing the phosphorylation of MYC. The levels of *miR-106b-93-25* cluster, which was downstream from *MYC*, were elevated, and this cluster inhibited the mitophagy-associated proteins (e.g., OPTN) to protect against excessive mitophagy, ultimately leading to cell death due to bioenergetic collapse [74,75]. This suggested that miRNAs targeting mitophagy-associated proteins were essential for cell survival and were a mechanism of mitophagy control.

A connecting piece of the puzzle is the link between mitophagy, autophagy, and a membrane protein. A study of *MYC*-driven lymphomagenesis found that Bax-interacting factor 1 (*Bif-1*), a member of the membrane curvature-driving endophilin protein family, also known as SH3GLB1, plays a role in apoptosis, autophagy, and mitophagy [76]. *Bif-1* was found to play a crucial role in the maturation of nascent autophagosomes during mitophagy to maintain chromosomal stability [76]. *Bif-1* haploinsufficiency in the Eμ-*MYC* mouse model (a widely used model of MYC-driven malignancy) accelerated *MYC*-driven lymphomagenesis by suppressing mitophagy and resulted in increased mitochondrial mass within cells at the premalignant state of *MYC*-induced lymphoma. Since the accumulation of damaged mitochondria (by suppressing autophagy) led to ROS generation and DNA damage accumulation, improper mitochondrial clearance was likely the mechanism behind the promotion of chromosomal instability induced by the loss of *Bif-1* in *MYC*-induced lymphoma [76].

In summary, in this section, we discussed the interplay between MYC regulatory networks with crucial cellular and molecular processes. We reviewed that MYC was involved in autophagosome formation. Moreover, MYC functional alterations in autophagic processes could contribute to tumourigenesis. We also revealed the role of multiple proteins affecting MYC protein stability. Finally, we investigated UPR, CMA, and mitophagy as cellular processes that may impact MYC functions and stability.

## 5. Discussion: Novel Treatment Options for MYC-Driven Malignancies in Light of the Topics Discussed

Chemotherapy has been instrumental in the treatment of MYC-driven cancers. However, a major caveat with these approaches is the non-discriminatory targeting of proliferating cells triggering widespread off-target effects. Thus, these methods are gradually replaced by more targeted and effective treatment modalities that rely on the specific targeting of factors that lead to the tumour’s initiation, propagation, and progression. One example of this is the notion of “oncogene addiction”, whereby, in theory, the genetic inhibition of oncogenes such as MYC is sufficient to cause cell death and reduced tumour burden. Additionally, due to the role of this oncogene in development, regulatable transgenic systems have been critical to these observations, allowing for a better understanding of the effects of de novo activation or inactivation of MYC [77]. For instance, in the context of preclinical mouse models, transcriptional activation of MYC in an inducible *c-MycER*™ mouse model of the epidermis resulted in papillomatosis, blood vessel formation, apoptosis, and precancerous lesions. Inversely, the lesions regressed upon tamoxifen removal and MYC inactivation [77]. This observation supported the idea that transient inactivation of an oncogene such as MYC can be considered a viable therapeutic strategy.

The regulation of MYC can also be achieved on a transcriptional level whereby the BET bromodomain protein, BRD4, interacts with the positive elongation factor complex b (p-TEFb), which is regulated by MYC [78]. Therefore, the BET bromodomain inhibitor, JQ1, was studied in multiple myeloma (MM) for its capacity to inhibit MYC-dependent transcription. In MM, the rearrangement of MYC via a translocation is a frequently occurring genetic alteration while the activation of MYC can be observed in over half of MM cases [79]. In a panel of MM cell lines, JQ1 treatment led to an increase in cells arrested in the G0/G1 phase of the cell cycle and cellular senescence (marked by B-galactosidase expression). Still, only modest levels of apoptosis were detected [79]. In an orthotopic mouse xenograft model, JQ1 treatment led to a decreased disease burden and increased overall survival of these mice compared to controls [79]. Furthermore, MYC-dependent processes such as glycolysis were also affected by JQ1. Collectively, these results pointed towards the feasibility of specifically targeting MYC and its extensive targets [79].

On a different note, MYC protein can also be modulated by various molecules [43,80]. For example, the interaction between lncRNA *LUCAT1* and Nuclein (NCL, an inhibitor of MYC) is interesting. *LUCAT1* interacted with NCL via its G-quadruplex structure (a structure formed by guanine rich nucleic acids) in CRC and promoted MYC expression and cancer cell proliferation. Interestingly, in knockdown and rescue experiments, cells lacking *LUCAT1* displayed reduced MYC levels, while this phenotype could be rescued by NCL knockdown [80].

Furthermore, it has proven possible to target MYC with morpholinos in lung and prostate cancers; the latter has been tested successfully in preclinical models and clinical trials [55,56]. Notably, morpholinos and other ASOs have been studied as successful approaches to target lncRNAs, some of which are regulated by or regulate MYC and can also be correlated with MYC via mutual modulation of expression and function [59,67,81]. Since lncRNAs can be upregulated under specific pathological conditions, the indirect targeting of MYC oncogenic functions via direct targeting of lncRNAs should, in principle, lead to improved treatment efficacy and reduced side effects. In this context, lncRNA-mediated upstream regulation of MYC should be further investigated, such as modulation of the *MALAT1/mir-204* axis in thyroid cancer, which, as reviewed in Section 3.1, could be a plausible method of reducing levels of MYC. We reviewed that *MALAT1* increased IGF2BP2 and MYC expression by binding to *mir-204* [32], a process that can be manipulated as a therapeutic strategy by inhibiting *MALAT1* or increasing *mir-204* levels. Other lncRNAs such as *linc00261, linc00485*, and *LUCAT1* may also show potential for regulating MYC levels and tumour-associated functions via their modulation [43,57,58].

MYC-driven gene regulatory networks also apply to the axis of miRNA and autophagy networks and, hence, may be exploited for treatment. For example, the involvement of *miR-27b-3p* in *MYC*-induced autophagy was investigated in CRC. In this study, CRC cells became resistant to oxaliplatin (OXA) through the *MYC*/*miR-27b-3p*/*ATG10* axis [82]. This study found that MYC, in association with relevant transcriptional machinery and repressors, repressed the transcription of *miR-27b-3p*, upregulating autophagy and leading to chemoresistance. Furthermore, *miR-27b-3p*, upon inhibiting autophagy, resensitised chemoresistant CRC cells to OXA through Atg10 suppression. Indeed, siRNA-mediated *ATG10* suppression inhibited the proliferation of SW480 colon adenocarcinoma cells and resensitised them to OXA [82]. Perhaps the most exciting finding of this study was the dissection of the *MYC*/*miR-27b-3p*/*ATG10* axis for therapy [82].

In line with the significance of various autophagic pathway components, including lysosomes in the completion of autophagy, lysosomal protein transmembrane 4 beta (LAPTM4B), a potential oncogene was investigated with respect to MYC regulatory networks in breast cancer [83]. For instance, LAPTM4B was shown to play a critical role in tumour proliferation; metastasis; autophagy inhibition; and resistance to chemotherapeutic drugs such as doxorubicin, cisplatin, and paclitaxel [83]. LAPTM4B-35 (the protein product) activated the PI3K/AKT signalling pathway leading to the phosphorylation of GSK3β, resulting in the attenuation of *MYC* degradation and consequently promoting the proliferation of carcinoma cells [84]. This significant finding suggested that LAPTM4B-35 may be an attractive target for reducing MYC levels. Additionally, the downregulation of *LAPTM4B-35* by RNAi decreased the efflux of chemotherapeutic agents, hence sensitising chemical therapy [84]. These studies have brought together MYC regulatory networks, lysosomal proteins, drug resistance, and autophagy as drug therapy candidates.

Finally, in addition to autophagy, the interaction of MYC with mitophagy and cellular metabolism was a focus of our review and may provide valuable biomarkers and therapeutic approaches in cancers. For instance, for diffuse large B-cell lymphoma (DLBCL), MYC induces abnormal choline metabolism by transcriptionally activating the essential gene, phosphate cytidylyltransferase 1 choline-α (*PCYT1A*), a gene involved in the synthesis of phosphatidylcholine [85]. The lipid-lowering alkaloid Berberine (BBR) exhibited anti-lymphoma activity by inhibiting *MYC*-driven *PCYT1A* expression and by activating mitophagy-dependent necroptosis, both in vitro and in vivo. B-cell lymphoma cell necroptosis was reversed by treatment with the mitophagy inhibitor Mdivi-1 [85]. This study revealed that the effective manipulation of MYC requires an understanding of the meshwork of cell death and survival, and autophagic and mitophagy processes.

In conclusion, the effective (in)direct targeting of MYC TFs through their regulatory networks with cellular and molecular networks in MYC-driven malignancies has shown promising potential for cancer therapy over traditional methods. In this review, we dissected numerous MYC networks associated with lncRNAs, autophagy, and mitophagy, in which specific targeting of MYC was achieved on transcriptional, epigenetic, miRNA regulation, or protein levels. These interactions and mutual regulations could be viewed as unexplored areas of MYC targeting that may warrant a more in-depth investigation with the objective of future clinical implementation. These strategies are in line with reducing off-target effects of non-specific and non-context-specific targeting of these TFs in the broader context of physiological and pathological networks and processes.

## Figures and Tables

**Figure 1 ijms-22-08527-f001:**
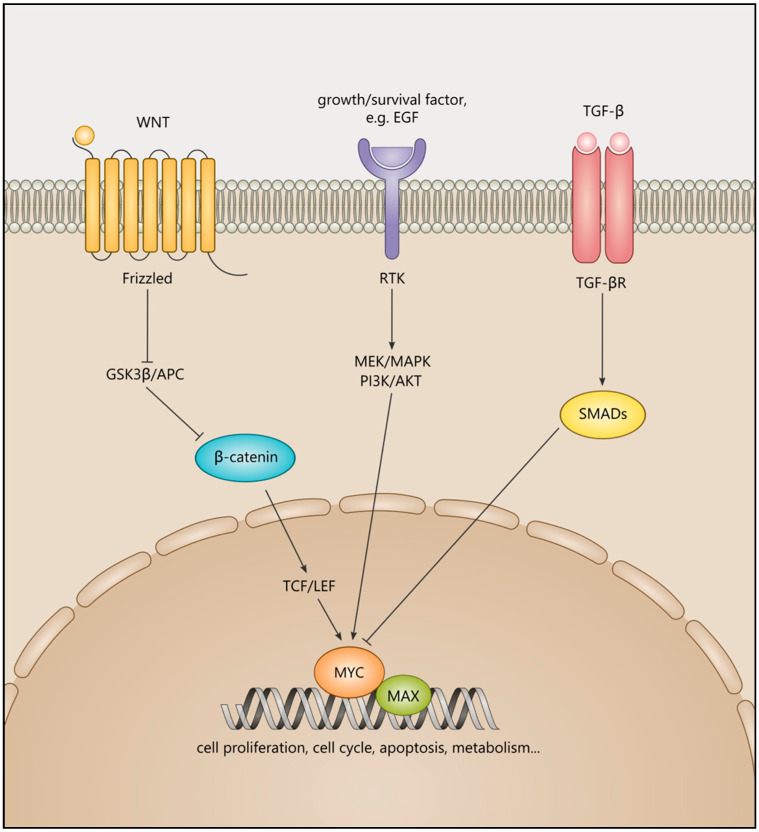
The effects of three signalling pathways converging on MYC activation or repression. A Wnt ligand binds to its receptor frizzled, preventing the phosphorylation of β-catenin by GSK3β and its subsequent degradation. β-catenin accumulates in the nucleus and, in association with T-cell factor/ lymphoid enhancer factor (TCF/LEF), activates Wnt signalling target genes, including *MYC*. The positive input of growth and survival factors including EGF and its downstream mediators, including MEK/MAPK/PI3K/AKT on MYC activation, while TGF-β signalling via SMADs can suppress MYC activity. MYC can form a heterodimer with MAX and can govern cell proliferation, apoptosis, and metabolism.

**Figure 2 ijms-22-08527-f002:**
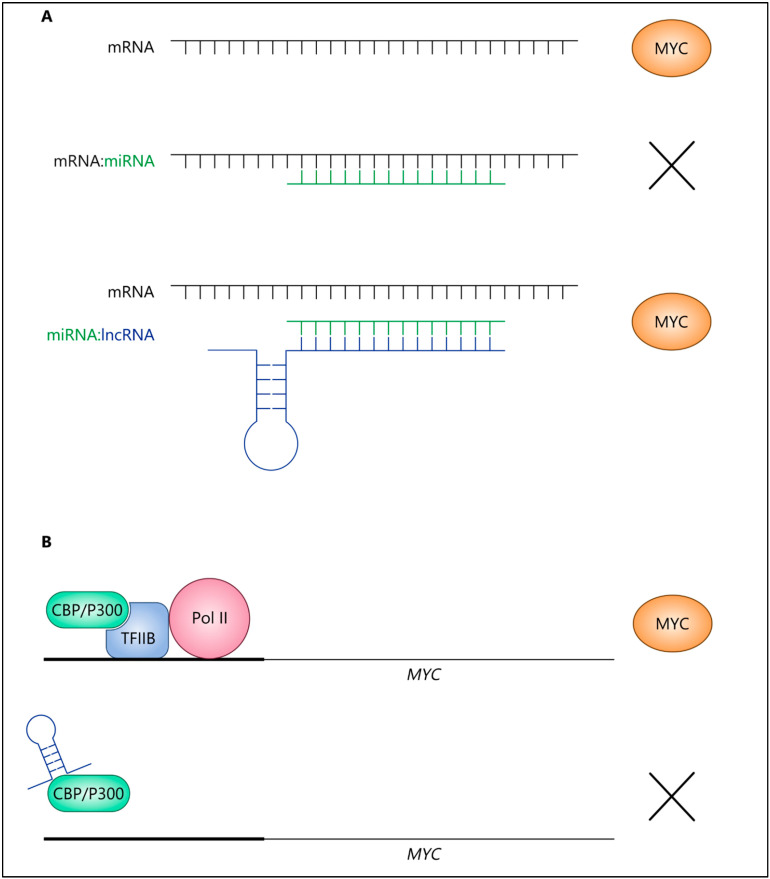
Regulatory loops involving lncRNAs and MYC. (**A**) A lncRNA (e.g., *MALAT1*) acting as a competing endogenous RNA by sequestering *miRNA* (e.g., *miR-204*), allowing for the production of MYC. (**B**) The physical interaction between a *lncRNA* (e.g., *linc00261*) prevents the recruitment of p300/CBP to the promoter region of *MYC*, thereby repressing *MYC* expression.

**Figure 3 ijms-22-08527-f003:**
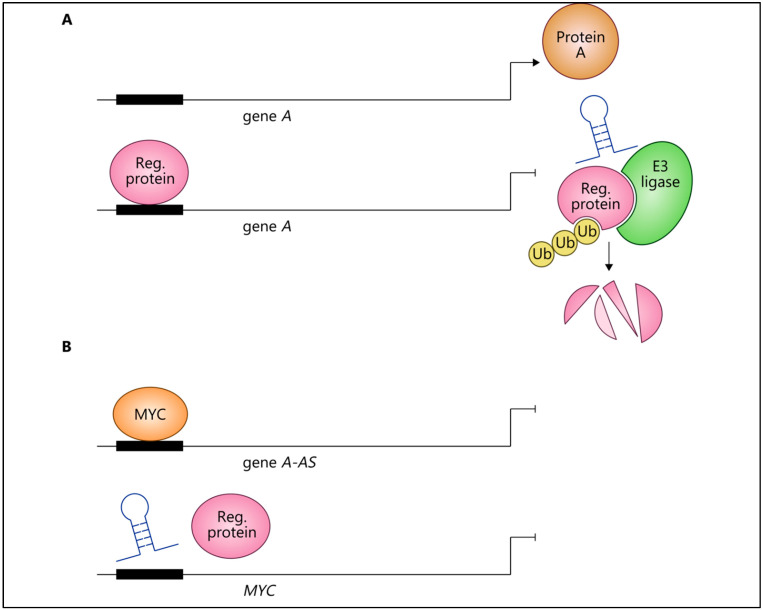
A negative feedback loop involving MYC, lncRNAs, and ubiquitin ligase. (**A**) *LncRNA* (encoded by gene *A-AS1*, e.g., *MTSS1-AS1*) upregulated the expression of *gene A* (e.g., *MTSS1* gene) acting as a scaffold between E3 ligase and a regulatory protein (e.g., MZF1), leading to ubiquitination-mediated degradation of the regulatory protein; hence, protein A (e.g., MTSS1) was produced. The regulatory protein (e.g., MZF1) inhibited *gene A* (e.g., *MTSS1* gene) expression by binding its promoter. (**B**) *MYC* inhibited the lncRNA (e.g., *MTSS1-AS1*) by binding its initiator elements (*gene A-AS*, e.g., *MTSS1-AS1*). In turn, the *lncRNA* (encoded by *gene A-AS*, e.g., *MTSS1-AS1*) inhibited MYC expression by impairing its regulatory protein-mediated transcriptional activation. Reg. = regulatory.

**Figure 4 ijms-22-08527-f004:**
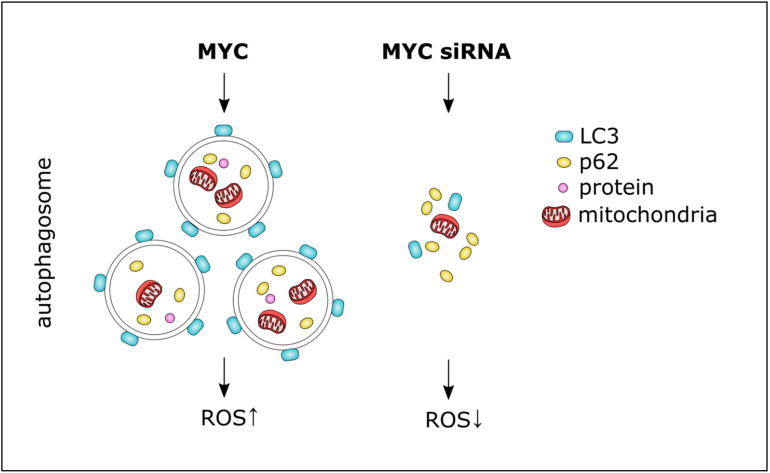
MYC and autophagy. MYC was involved in autophagosome formation whereby siRNA-mediated knockdown of *MYC* inhibited autophagy, resulting in the accumulation of the autophagy substrate p62. MYC is known to trigger ROS accumulation.

**Figure 5 ijms-22-08527-f005:**
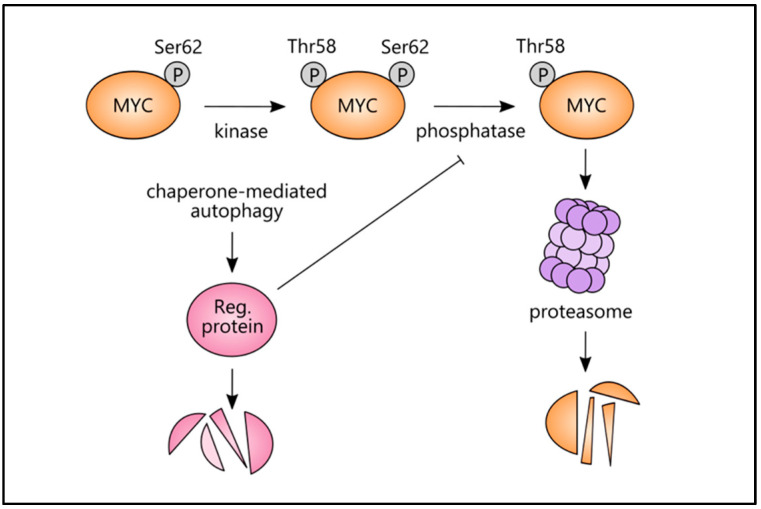
CMA and MYC protein stability. A phosphatase removes phosphoserine 62 from MYC, rendering it unstable and susceptible to proteasomal degradation. This phosphatase may be suppressed by a regulatory protein (e.g., CIP2A) that can, in turn, be degraded by CMA-associated processes. Reg. = regulatory.

**Table 1 ijms-22-08527-t001:** Examples of MYC-associated partners, upstream mediators, or target genes and the malignancies in which they have been identified.

Cancer Type	Examples of MYCN/MYC/MYCL-Associated Binding Partners, Upstream Regulators, Effectors, and Target Genes and Their Signalling Pathways
Prostate cancer	Co-occupation with AR at *FOXA1* and *HOXB13* loci [24]
Pancreatic ductal adenocarcinoma	KRAS and YAP [25]
Medulloblastoma	SOX2 [26]
Neuroblastoma	Wnt/ β-catenin, *mir-204* [27,28]
Colorectal cancer	Wnt-APC, *CAD*, *UMPS*, and *CTPS* [7]
Rhabdomyosarcoma	PAX3-FOXO1 [29]
Non-small cell lung cancer	KRAS, IL-23, and CCL9 [30,31]

**Table 2 ijms-22-08527-t002:** Strategies to modulate MYC-driven gene networks concerning regulatory, lncRNAs, and autophagy-related processes and networks.

Target	Mechanism of Regulation	Outcome	Malignancy
MYC	Antisense oligonucleotide morpholinos	Reduction of pro-angiogenic proteins and metastasis	Lung cancer [55]
*linc00485*/*miR-298*/MYC network	*linc00485* silencing	Reduction of cancer cell proliferation	Lung cancer [57]
*Linc00261*/p300/CBP/MYC network	*Linc00261* overexpression	Inhibition of cancer cell proliferation, migration, and metastasis in vitro	Pancreatic cancer [58]
*LPP-AS2/miR-7-5p*/EGFR/MYC	*LPP-AS2* knockdown	Tumour growth inhibition	Glioma [59]
*MYC*/*miR-150*/*EPG5* (autophagy)	Inhibition of *MYC/miR-150* expression	Reduced cell growth in vitro and in vivo	NSCLC [60]
AMBRA1-PP2A/MYC regulated by mTOR	mTOR inhibition	MYC protein degradation and reduction in cell proliferation	Breast and lung carcinoma cell lines [61]
MYC activation of PERK/eIF2α/ATF4 arm of the UPR	PERK inhibition Autophagy inhibition	Reduced MYC-induced autophagy and tumour formation, Increase in MYC-dependent apoptosis	Human P493-6B lymphoblastoid cell line and MEF cells [62]
CMA/ CIP2A/ MYC	CMA activation	Proteasomal degradation of MYC	MEF cells [63]

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
