# Peer review of "The Contribution of Autophagy and LncRNAs to MYC-Driven Gene Regulatory Networks in Cancers"

_ijms, 2021, doi:10.3390/ijms22168527_

Round 1

Reviewer 1 Report

The paper is well-presented and reviewed in a proper way. Autophagy is a complicated field and the authors provived all the necessary informations to understand the topic. The link between this process, lncRNAs and MYC is obviously interesting due to the possibilities in terms of new therapeutic strategies. 

Author Response

We would like to thank the reviewer for the comments.

Reviewer 2 Report

The review by Jahangiri et al. discusses the role of the oncoprotein Myc in concerting oncogenic networks in various cancers. The authors focuses on several aspects of Myc biology, i.e., its particular functions in different cancer types (chapter 2), its connection to lncRNAs (chapter 3), its interplay with autophagy (chapter 4), and therapeutic strategies to target Myc in cancer (chapter 5). All of these aspects are interesting and very relevant for our understanding of the plethora of ways in which Myc drives tumorigenesis. However, given the complexity of each aspect on its own, I feel that the authors have been too ambitious to combine them all in one manuscript. In fact, each chapter appears to rush through the topic, failing to give a clear and concise overview of the topic. Some parts have too much experimental details, e.g., when it comes to the description of mouse models like on pages 4 and 7, whereas very often the abbreviations of factors, genes and so on are not properly introduced. Moreover, there are also some logical flaws, that make it hard to understand the text. Also the choice and design of the figures is not too fortunate. Figure 1 does not really fit into the context of the review (signalling towards Myc is not a major point addressed) and the other figures are of poor quality and kept in most cases generic (e.g., “Reg.protein”, “Protein A”), which does not contribute to a better understanding of the text. Based on these conceptual concerns and the individual critiques listed below, I suggest a major revision before the manuscript can be re-evaluated for publication. In particular, the manuscript would benefit from reducing the number of topics addressed to one or two, and to discuss these topics adequately.

Individual critiques:

  • Please introduce the following abbreviations (e.g. APC, TEAD, SHH, NSG, NCL, Lin 28B) with their full-length name. In some cases, also a short additional explanation would be helpful.
  • page 3, line 77: “stem-related genes” should be “stem cell-related genes”
  • page 4, lines 134-146: Here, and in many other sections of the text, the writing is in italics. Why is this the case?
  • page 5, lines 174-175: “YAP in partnership with TEAD TFs directly transcribed Myc...”. How can TFs transcribed a template? A clear distinction between the Myc protein and gene would also be helpful.
  • page 6, line 255: The “withdrawal of doxycyline” is mentioned, however, the system in which system the drug is used is not properly explained.
  • page 7, line 298: Please correct “approxima1tely”.
  • page 8, line 322: The URL is not properly embedded in the text.
  • page 8, lines 334-340: The discussion of morpholinos against Myc mRNA does not really fit to this chapter concerning Myc and lncRNAs.
  • page 8, lines 349-350: The statement that “MALAT1 recognises m6A modification” does not make sense. The authors should revisit the original paper.
  • page 9, line 372: The statement that linc00261 acts “via its interaction with MYC” is not clear and does not fit to Figure 2B. Does the lncRNA rather bind to CBP/p300 and maybe to the Myc gene promoter? The authors need to be clarify this statement.
  • page 10, lines 384-385: “Linc00261 is downregulated by hypermethylation of its promoter mediated by EZH2...” EZH2 does not mediate promoter (i.e., DNA) methylation. The authors should be more precise.
  • page 10, lines 386-389: “Interestingly, linc00261 downregulation correlates with advanced tumour stage and poor prognosis. in vivo, linc00261 is downregulated in tumour tissues which correlates with an improved patient prognosis”. Both statements contradict each other and thus need clarification.
  • page 10, lines 300-401: “PVT1 lncRNA is associated with a poor prognosis for 400 CRC patients.” Shouldn’t PVT1(b), as an inhibitor of Myc expression, been associated with good prognosis?
  • page 10, lines 402-417, and Figure 3: The explanation of the feedback-loop is not clear, and the text and Figure 3 are not consistent. MZF1, which is “Reg. protein” in the figure, is a repressor of MTSS1 (“gene A”). MTSS1-AS induces degradation of MZF1, thus sustaining MTSS1 But does MTSS1-AS really bind to the Myc gene promoter as shown in the figure? Or does it inhibit Myc expression by reducing the levels of MZF1, which in turn is an Myc activator? This interplay needs to be clarified.
  • page 11, line 426: “miRNA that inhibits the transcription...” I guess the authors meant translation?

Author Response

The review by Jahangiri et al. discusses the role of the oncoprotein Myc in concerting oncogenic networks in various cancers. The authors focuses on several aspects of Myc biology, i.e., its particular functions in different cancer types (chapter 2), its connection to lncRNAs (chapter 3), its interplay with autophagy (chapter 4), and therapeutic strategies to target Myc in cancer (chapter 5). All of these aspects are interesting and very relevant for our understanding of the plethora of ways in which Myc drives tumorigenesis.

We thank the reviewer for this comment.

However, given the complexity of each aspect on its own, I feel that the authors have been too ambitious to combine them all in one manuscript. In fact, each chapter appears to rush through the topic, failing to give a clear and concise overview of the topic. Some parts have too much experimental details, e.g., when it comes to the description of mouse models like on pages 4 and 7, whereas very often the abbreviations of factors, genes and so on are not properly introduced. Moreover, there are also some logical flaws, that make it hard to understand the text. Also the choice and design of the figures is not too fortunate. Figure 1 does not really fit into the context of the review (signalling towards Myc is not a major point addressed) and the other figures are of poor quality and kept in most cases generic (e.g., “Reg.protein”, “Protein A”), which does not contribute to a better understanding of the text. Based on these conceptual concerns and the individual critiques listed below, I suggest a major revision before the manuscript can be re-evaluated for publication. In particular, the manuscript would benefit from reducing the number of topics addressed to one or two, and to discuss these topics adequately.

-We thank the reviewer for their constructive comments. Sections 1 and 5 are introduction and discussion that are fundamental to the manuscript and we think would affect the quality of the review.  With respect to the 3 main topics covered (MYC in cancers, lncRNA, and autophagy), we had previously checked and obtained editorial permission for the topics covered since this was an invited review. After discussing these deletions at length with the authors we thought of the following:

1- Sections 3 and 4 comprise our core messages that have been mentioned in the title of the paper, and the deletion of both or either of these sections, will completely alter our paper and core message.

2- The respected reviewer has kindly provided some points and suggested minor editing for sections 2 and 3. Hence, deletions of these two sections would prevent us from addressing these comments.

3- Deletions of sections 2,3 and 4, will lead to further significant deletions. For instance, deletion of section 2 (MYC in cancers), would also lead to the deletion of table 1 that summarises section 2. If we retain table 1, it will appear as an unexplainable stand-alone insertion. On a similar note, deletion of section 3 (lncRNA), would not only alter the identity of the paper but would lead to the deletion of figures 2 and 3, parts of table 2, and large sections of the discussion (section 5) regarding lncRNAs. Finally, deletion of section 4, would lead to the deletion of figures 4 and 5, parts of table 2, and large sections of the discussion concerning autophagy.

In light of these significant issues raised, we propose the following resolution:

1- We have comprehensively reviewed the manuscript to provide more detail throughout in relevant sections (shown in blue).  In addition, at the end of each section, we have provided a paragraph explaining the content of the section in a clear and precise way. In total, we have added 2-3 pages of more detailed explanation of the existing topics, without adding any new articles.

2- With respect to the logical flows, we have reviewed the flow thoroughly across the manuscript and made modifications to improve it.

3- We have referred to the mouse experiments on pages 4 and 7 and omitted detailed experimental information as requested (please refer to the manuscript).

4- We have also asked all the authors, including native English speakers to read and edit the manuscript to improve the readability as requested.

5- With respect to the figures, we have dramatically improved the quality of figures 2-5. We were, however, very concerned about using actual protein/ lncRNA/ gene names in these figures and instead used generic terms. Reproducing figures from published papers, still utilises the original concept and ideas of the article and requires explicit permission from the original authors, and failing to do so would be viewed as an infringement of copyright rules and plagiarism, hence the practice of using generic terms. As a resolution, we have provided the actual names in brackets as examples of the proposed mechanisms to both steer clear of copyright violation and also to be as precise and clear as possible.

6- We have reviewed figure 1, we fully agree with the reviewer that upstream signalling of MYC was not the main focus of the paper, however, we feel that this figure, as it stands, will allow the readers,  to refer to it and refresh their memory when reading about these pathways and MYC throughout the manuscript. Also, we have referred to these pathways on numerous occasions (please see below) and a visual representation of the pathway will improve the readability.

- Wnt in lines 61-69, multiple cancer types (medulloblastoma, neuroblastoma, and colorectal cancer), and 430-434

- PI3K/AKT signalling in lines 62-70, lines 484-486, 496-500, 680-684, 736-739

- TGF-β/ SMADs in lines 61-69 and 430-434

We believe that these major revisions have substantially improved our manuscript while preserving the integrity of our message and sincerely hope that the reviewer also agrees with us and is satisfied with the changes made.

Individual critiques:

  • Please introduce the following abbreviations (e.g. APC, TEAD, SHH, NSG, NCL, Lin 28B) with their full-length name. In some cases, also a short additional explanation would be helpful.

We thank the reviewer for this constructive comment and have provided abbreviations for the mentioned terms.

  • page 3, line 77: “stem-related genes” should be “stem cell-related genes”

We thank the reviewer for this constructive comment and have corrected this error.

  • page 4, lines 134-146: Here, and in many other sections of the text, the writing is in italics. Why is this the case?

We are not sure when this anomaly took place, it is possible that changes have taken place during typesetting since we have not found this in our original manuscript. Nonetheless, we have addressed this anomaly and similar anomalies across the paper.

  • page 5, lines 174-175: “YAP in partnership with TEAD TFs directly transcribed Myc...”. How can TFs transcribe a template? A clear distinction between the Myc protein and gene would also be helpful.

We thank the reviewer for this comment and have now made a distinction between the Myc gene and protein.

  • page 6, line 255: The “withdrawal of doxycycline” is mentioned, however, the system in which system the drug is used is not properly explained.

We have provided more detail about this. PAX3-FOXO1 was doxycycline-inducible while MYCN was constitutively active. Mice transplanted with both MYCN and PAX3-FOXO1 expressing cells (+doxycycline induction) generated tumours, while PAX3-FOXO1 alone, led to tumours in a longer period of time. Doxycycline induction was achieved by adding this drug as a supplement to the diet of relevant mice. We have included this explanation in the relevant section in the manuscript.

  • page 7, line 298: Please correct “approxima1tely”.

We have corrected this error.

  • page 8, line 322: The URL is not properly embedded in the text.

We have checked the URL and it works in our word document. We will discuss this with our allocated assistant editor since we cannot pinpoint the issue.

  • page 8, lines 334-340: The discussion of morpholinos against Myc mRNA does not really fit to this chapter concerning Myc and lncRNAs.

We have omitted this section.

  • page 8, lines 349-350: The statement that “MALAT1 recognises m6A modification” does not make sense. The authors should revisit the original paper.

We have referred to the original paper and modified this section.

  • page 9, line 372: The statement that linc00261 acts “via its interaction with MYC” is not clear and does not fit to Figure 2B. Does the lncRNA rather bind to CBP/p300 and maybe to the Myc gene promoter? The authors need to be clarify this statement.

Based on in silico predictions and RNA immunoprecipitation (RIP), conducted by the original paper, the authors have revealed that linc00261 can bind to the acetylase p300/CBP, but not to other HDACs.

  • page 10, lines 384-385: “Linc00261 is downregulated by hypermethylation of its promoter mediated by EZH2...” EZH2 does not mediate promoter (i.e., DNA) methylation. The authors should be more precise.

We have reviewed this link more precisely. The downregulation of linc00261 is achieved by the methylation of the CpG islands associated with its promoter, while the methylation of H3K27 sites located in the promoter regions of this lncRNA is mediated by EZH2. We have added this to the manuscript.

  • page 10, lines 386-389: “Interestingly, linc00261 downregulation correlates with advanced tumour stage and poor prognosis. in vivolinc00261 is downregulated in tumour tissues which correlates with an improved patient prognosis”. Both statements contradict each other and thus need clarification.

Linc00261 was significantly downregulated in PC tissues, and its expression was positively associated with the prognosis of PC patients”

We have referred to the original paper and have decided to only include the sentence above.

  • page 10, lines 300-401: “PVT1 lncRNA is associated with a poor prognosis for 400 CRC patients.” Shouldn’t PVT1(b), as an inhibitor of Myc expression, been associated with good prognosis?

We have moved this sentence to after the PVT1 section since it accurately describes PVT1 properties and not PVT1b.

  • page 10, lines 402-417, and Figure 3: The explanation of the feedback-loop is not clear, and the text and Figure 3 are not consistent. MZF1, which is “Reg. protein” in the figure, is a repressor of MTSS1 (“gene A”). MTSS1-AS induces degradation of MZF1, thus sustaining MTSS1 But does MTSS1-AS really bind to the Myc gene promoter as shown in the figure? Or does it inhibit Mycexpression by reducing the levels of MZF1, which in turn is an Myc activator? This interplay needs to be clarified.

We have provided examples of gene, protein, and lncRNA names, to both avoid the infringement of copyright rules but to better explain the figure. Upon referral to the original paper, the content has been confirmed. MTSS1-AS repressed the expression of MYC by impairing MZF1 (reg protein)-mediated transcription activation of this gene. Specifically, the authors used ChIP to show that the knockdown of MTSS1-AS led to increased binding of MZF1 (reg protein) to MYC promoter (while the overexpression of MTSS1-AS inhibited MZF1 binding to MYC promoter). Refering to the figure, the lncRNA is now not binding the promoter but preventing MZF1 (reg protein) from binding (the previous design of the lncRNA may have given that impression because of its shape). We fear that if we move it up any further, it might not show blockage of the reg protein binding to the promoter.

  • page 11, line 426: “miRNA that inhibits the transcription...” I guess the authors meant translation?

       We have corrected this mistake, thank you sincerely  for your guidance.

Round 2

Reviewer 2 Report

In the revised manuscript, the authors have added quite some text, instead of shortening the paper as suggested.  Unfortunately, this lengthening has not aided the readability and the paper is even more overloaded with information. Several logical flaws (not flows!) have still not been resolved, e.g.:

  • Abbreviations are still not introduced properly when used the first time (it's not really helpful to have them just spelled out in the list of abbreviations).
  • Transcription factors can still not transcribe a gene (they can only direct the RNA polymerase).
  • The URL is still not properly embedded in the text (even if it is functional). You should make clear how this URL relates to your text.
  • What are H3K27 sites? This term is uncommon. Spelling out the abbreviation would help readers who are not so familiar with epigenetics.

In summary, I think the revised version has become longer, but not significantly improved.

Author Response

In the revised manuscript, the authors have added quite some text, instead of shortening the paper as suggested.  Unfortunately, this lengthening has not aided the readability and the paper is even more overloaded with information. Several logical flaws (not flows!) have still not been resolved, e.g.:

We thank the reviewer for pointing out these oversights. In order to reduce the length of the manuscript, we have merged or deleted sentences, and also removed whole paragraphs across the manuscript. These edits reduced the length while preserving the detailed explanation we have provided.

For instance:

1- We have deleted the role of MYC in somatic reporgramming and other sentences from the introduction;

2- We have deleted the mention of mitophagy from the adenocarcinoma section in section 2 along with some other deletions that have helped to reduce the information overload;

3- We have limited our cancer type reviews to solid cancers only, hence T-ALL relevant information was removed, including from table 1;

4- From the lncRNAs section we have removed the XLC_006390 (3.2) and linc00839 (3.4) paragraphs;

5- From the autophagy section, we have removed the section on 4-O-methyl-ascochlorin (4.1), in addition to some other sections (please refer to the maint text).

6- From the discussion, we have deleted the G9a and AM879  information, since epigenetics was not a major focus of the paper. In addition, we have deleted the section on caski cells.

In total, we have deleted over 5 pages (around 20% less content).

Again, we have conducted language editing.

  • Abbreviations are still not introduced properly when used the first time (it's not really helpful to have them just spelled out in the list of abbreviations). We have gone through the manuscript to check for any gene/protein in focus that was not introduced and have provided more detail (text in blue).
  • Transcription factors can still not transcribe a gene (they can only direct the RNA polymerase). We have searched for any issues with the description of transcription, although we do not mean that a TF can individually transcribe a gene and completely agree with the reviewer that the role of the transcription machinery plays a vital role. Hence, in multiple locations, we have acknowledged the role of the machinery to remind the readers of this important point. We also have altered the language in many locations to read “promote” or “influence” transcription.
  • The URL is still not properly embedded in the text (even if it is functional). You should make clear how this URL relates to your text. We have addressed the URL issue and also provided a more specific explanation about it.
  • What are H3K27 sites? This term is uncommon. Spelling out the abbreviation would help readers who are not so familiar with epigenetics.

We have introduced H3K27 and all other abbreviations.

In summary, I think the revised version has become longer, but not significantly improved.

We believe that these substantial changes have improved the readability of the manuscript and decreased the information overload, and sincerely hope that the reviewer is satisfied with our changes.

Round 3

Reviewer 2 Report

The efforts to make the manuscript more succinct are appreciated. The manuscript has improved, publication is justifiable.